# Development and Validation of a LabVIEW Automated Software System for Displacement and Dynamic Modal Parameters Analysis Purposes

**Reina El Dahr \*, Xenofon Lignos, Spyridon Papavieros and Ioannis Vayas**

Institute of Steel Structures, National Technical University of Athens, 15772 Athens, Greece; vastahl@central.ntua.gr (I.V.)
\* Correspondence: rdahr@mail.ntua.gr

**Abstract:** The structural health monitoring (SHM) technique is a highly competent operative process dedicated to improving the resilience of an infrastructure by evaluating its system state. SHM is performed to identify any modification in the dynamic properties of an infrastructure by evaluating the acceleration, natural frequencies, and damping ratios. Apart from the vibrational measurements, SHM is employed to assess the displacement. Consequently, sensors are mounted on the investigated framework aiming to collect frequent readings at regularly spaced time intervals during and after being induced. In this study, a LabVIEW program was developed for vibrational monitoring and system evaluation. In a case study reported herein, it calculates the natural frequencies as well as the damping and displacement parameters of a cantilever steel beam after being subjected to excitation at its free end. For that purpose, a Bridge Diagnostic Inc. (BDI) accelerometer and a displacement transducer were parallelly mounted on the free end of the beam. The developed program was capable of detecting the eigenfrequencies, the damping properties, and the displacements from the acceleration data. The evaluated parameters were estimated with the ARTeMIS modal analysis software for comparison purposes. The reported response confirmed that the proposed system strongly conducted the desired performance as it successfully identified the system state and modal parameters.

**Keywords:** structural health monitoring; damping; eigenfrequency; displacement; LabVIEW; ARTeMIS

## 1. Introduction

SHM is a highly competent operative process dedicated to improving the resilience of an infrastructure by evaluating its system state to validate numerical models and to approve its safety standards by spotting structural abnormalities or damages. Therefore, employing structural assessment methodologies in the civil engineering sector is deemed to require a significant performance, attempting to find the infrastructure sustainability for upcoming growth [1,2]. SHM is intended to provide a non-destructive assessment of the structural conditions by means of evaluating the dynamic properties and investigating any modification found in the structural behavior [3] at any time during its life span. Accelerations, natural frequencies, and damping ratios are regarded as crucial system properties to measure or assess in order to characterize the structural state [4] and investigate its response to any inducement it may be subjected to during its operative working life [5].

Along with vibrational parameters, a displacement calculation is significant as it is implemented to validate numerical models, determine the dynamic features of a structure [6], and assess the structural deterioration that the framework is subjected to [7]. The structural performance of a civil engineering infrastructure may be analyzed accordingly on the basis of the evaluated modal parameters along with in-time measurements. Such a procedure facilitates the maintenance plan by prioritizing more damaged structures in the aim of restoring and fixing them [8].

Operational modal analysis (OMA) in the field of health monitoring is adopted to qualify the system condition and identify any changes in the vibrational structural parameters [3,9,10]. This consists of converting the reported oscillation signals of a stimulated system into a collection of parameters that are conveniently evaluated. Thus, sensing detectors are mounted on the investigated framework in the aim of collecting frequent readings at regularly spaced time intervals for the purpose of identifying unusual behavior that can threaten the wellbeing of the structure [11].

## 2. Natural Frequency Calculations

The flowchart of Figure 1 illustrates the procedure involved in processing measurements recorded by accelerometers in order to calculate the natural frequencies:

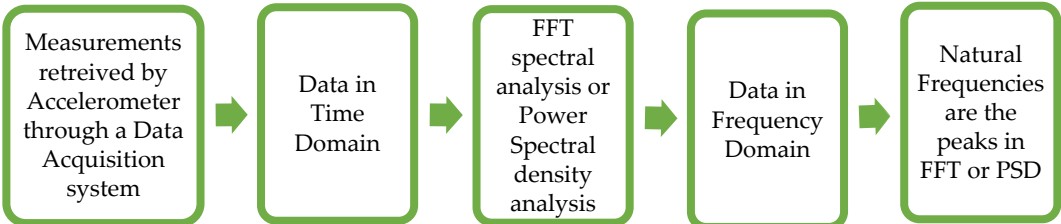

**Figure 1.** Flowchart of the signal processing procedure for a natural frequency calculation.

When stimulating a structural component by a constant force, with different frequency rates, the member denotes a more forceful response as the rate of vibration of the applied force becomes closer to the system's natural frequency, at which it reaches its optimal reaction [12]. The reported dynamic oscillations can be represented either in time or in the frequency domain [13].

The installed sensors report data in the time domain. The recorded signal is then subjected to a fast Fourier transform (FFT) in order to convert the data into the frequency domain (Figure 1). The fast Fourier transform technique allows the calculation of a power spectral density by applying the hamming spectral window operator [14]. The calculation of the frequency domain reveals high peaks, where each peak refers to one of the system's natural frequencies [12].

## 3. Damping Ratio Calculations

Besides the natural frequencies, OMA allows the damping of the ratio calculation. A structure typically stores energy after being stimulated. Damping is the pace at which this energy is gradually released, resulting in a reduction in the magnitude of an unrestrained vibrational process [15]. However, damping property calculations are more challenging compared to the natural frequency. Ciornei in 2009 adopted the logarithmic decrement in displacement to measure the damming of a wooden beam, clipped from one end and free from the second end. It was stimulated by an initial displacement. The logarithmic decline of the measured reaction was used to compute the structural damping. This research yielded accurate dynamic properties [16].

The damping ratios of a studied structure are associated with the natural frequencies. Therefore, it is necessary to apply a bandwidth filter with cutoff frequencies surrounding the eigenfrequency in order to calculate the relevant damping ratio (Figure 2).

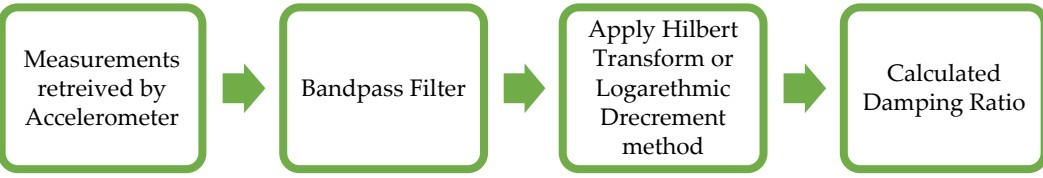

**Figure 2.** Flowchart of the signal processing procedure for the damping ratio calculation.

The flowchart of Figure 2 illustrates the procedure involved in the damping ratio calculation when processing measurements recorded by the accelerometer.

## 4. Displacement Parameter Calculations

In addition to the calculation of the vibrational parameters, displacement quantification is required to fully understand the structural behavior for health monitoring purposes. While dynamic assessments are carried out in order to examine the infrastructure's dynamic properties, static evaluation is performed to determine the structural stiffness and load capacity [17] as they are implemented in seismic design [18], deterioration assessment [19,20], system identification purposes [21,22], and structural health monitoring [23–26].

Despite their vital relevance in terms of being among the most helpful indicators of an infrastructure's health state, the application of sensors to calculating displacements is limited [27]. This is due to the fact that many structures are located on steep terrains where it is challenging to mount conventional displacement sensor networks, such as linear variable differential transducers (LVDTs), in the absence of fixed reference points [28–30].

In order to evaluate the displacements, numerous methods can be considered that are divided into connected and non-connected systems. Among the connected ones, we can cite LVDTs, global positioning systems (GPSs) [31–33], and non-direct estimated calculation using acceleration and velocity or strain-recorded data [34–39], whereas among the non-connected, we can list the laser Doppler vibrometers (LDVs) [30,40], vision-based schemes [41,42], or the total station [43].

Not only LVDTs but also all non-connected methodologies require a fixed reference [44–46]. They offer precise evaluation but are typically unviable as they are so expensive and difficult to implement [47].

Laser Doppler vibrometers (LDVs) do not necessitate any real interaction with the intended point to measure, so these are gaining prominence [22,48]. However, their precision can be restricted by any natural phenomenon that does not ensure a clear atmosphere enabling the direct visibility of the measured point. For example, the precision of the results may be affected by bad or low-visibility weather, or if the location of the point is concealed and the sensor cannot catch it. Moreover, LDVs are considered expensive compared to the BDI accelerometer. Therefore, they are not suitable for long-term monitoring to keep them on site. Although LDV is fairly precise when measuring parameters, it is only implemented for short-range displacement. Furthermore, a single LDV can detect the displacement of a single spot, which is considered inefficient [49,50].

The global positioning system (GPS) [51–54] is one of superior options. The GPS implementation does not require any reference marks, however, due to its high cost and relatively low precision, a displacement assessment can be performed at a predefined key location where the calculated dislocation is large enough for the GPS to detect it. Generally, GPS is used for surveying purposes [55,56], but it is not sufficiently accurate to be adopted for displacement evaluation in the civil engineering field.

Structural displacement measurements may also be performed by adopting vision-based approaches [57,58]. However, obstacles persist in the present system, including a custom constructed optical apparatus [57] or treating complex signals and analyzing them off-line [58]. Vision-based assessment was recently adopted with affordable cameras to measure displacements [59–61]. However, due to bad vision, its implementation is not recommended since it influences the precision of the results. Lee and Shinozuka in 2006, presented a vision-based method for measuring the dynamic displacement of the Yeondae bridge with two steel girders by adopting an authentic image processor. This approach is inexpensive and simple to use, as it can measure dynamic displacements with great accuracy [27]. The displacement results were close to the laser vibrometer carrying small noise. However, environmental conditions such as wind can shake the camera and cloudy weather may lead to inaccurate results that lower the precision of the captured images due to missing light sources. Therefore, it is proposed for financial and technical aspects to

quantify both the accelerations and displacements by implementing accelerometers as it is effective without the need for reference benchmarks [35,62].

Due to the accelerometers' ease of handling and set up, as well as minimal recorded noise, researchers have considered indirect estimating techniques to translate the acceleration to displacement as a useful option to overcome all the difficulties imposed by displacement sensors [28]. Both velocity and displacement measurements can be calculated by integrating acceleration data collected by accelerometer sensors mounted on large and complex civil engineering structures, making it the most efficient and least expensive technique [63,64]. It is feasible to calculate the displacement versus time by doubly integrating the acceleration–time history as it is simple to mount accelerometers with a low-cost sensor network. In a study carried out in 2017, Sekiya et al. adopted 10 MEMS accelerometers to calculate the displacement of a bridge. The suggested methodology consists of applying a high-pass filter (above 1 Hz) on the Fourier transform of the forced displacement part of the acceleration, and then apply inverse FT to integrate the acceleration data and subtract the drift from velocity after first integration and from the displacement at the second integration [65]. Recorded acceleration data may contain either low frequency noise or both. Chiu in 1997 [66] described in his research that low-frequency noise may influence the baseline and cause errors. This is due to sensors' noise or the environmental background. Therefore, the most essential obstacle to overcome in order to adopt an acceleration double integration strategy is to reduce the inaccuracies induced by the calculations [67].

A high-pass filter is used in order to exclude low frequencies and noises, so that a clear dynamic reaction is properly predicted from the acceleration [29].

The flowchart of Figure 3 illustrates the procedure involved in the displacement calculation by integrating acceleration data.

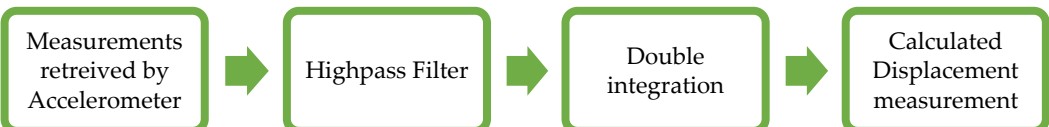

**Figure 3.** Flowchart of a signal processing procedure for the displacement measurement calculation.

Experimental modal testing technique is a non-destructive assessment approach frequently based on the excitation of a hammer: it defines the aspect of any structure with regard to its dynamic qualities such as natural frequency, damping, and mode shapes [4,15]. In 2015, W. Prashant et al. performed an analytical and experimental modal testing two distinct metals constituting a cantilever beam perturbed by a hammer stimulation. The analysis result included the analysis of the natural frequency, damping. and mode shapes. The NI LabView software was implemented to generate the frequency response components and then calculate the modal parameters. Both the theoretical and real results show strong similarity with a respectable margin of error [15].

In this study, a steel cantilever beam subjected to an operational and experimental modal testing process involving excitations at its free end was tested in order to evaluate its vibrational measurements and calculate its displacement. Accelerometers were installed to record acceleration data, which were later used to calculate natural frequencies, damping ratio, and displacement. In addition, a displacement transducer was used to measure the displacement and validate the analysis procedure. Displacement was calculated by the double integration of the recorded acceleration data, and a high-pass filter was implemented (Figure 3). Both sensors were connected to a DAQ system developed by El Dahr et al. in 2022. It is based on a digital processor as it incorporates the collection and evaluation of the measurement data [68]. The DAQ system comprises sensors to collect data and converts it from physical values into electric waveform, an acquisition hardware component based on the digital computer [69] to process the recorded signal and encode it from the signal to numerical data as well as a server to receive the data and calculate the output through a programming software: LabVIEW.

A comparative analysis between the newly developed LabVIEW code and the ARTeMIS modal standard version was performed to check the validity and effectiveness of the suggested coding language.

Both programs were adopted to assess large projects. For instance, LabVIEW was implemented by Li et al. in 2006 to develop the code to gather the signals recorded by sensors mounted on the Shandong Binzhou Yellow River highway bridge located in China for structural health monitoring purposes [70]. The same was observed for ARTeMIS, where the ambient assessment of the Vasco da Gama bridge located in Portugal was analyzed in ARTeMIS during the design validation stage. Moreover, it was adopted by projects monitoring smaller bridges, such as the S101 highway bridge in Austria, to assess the damage and degradation over time due to environmental influences [71].

The Laboratory Virtual Instrument Engineering Workbench (LabVIEW) platform is a programming software that operates via a graphical interface, primarily adopted for data acquisition and structural assessment purposes. The most notable advantage of this technique is that it supports a defined conceptual analysis. It facilitates and ensures efficient employment in different varieties of projects such as bridges, towers, and high-rise buildings, while it shortens the computational burden by adjusting minor inputs to finally detect modal properties.

Moreover, it overperforms different DAQ techniques considering its accuracy in collecting data and the analysis strategy, owing to its topology design and the convenient network of used sensors.

## 5. Experimental Benchmark Procedure

The test specimen was a cantilever beam with a steel plate cross-section of 80 mm height × 6 mm thickness and a total length of 1.86 m (Figure 4). Sensors were mounted on the edge of the beam and measurements were taken upon perturbation. Excitations were generated by arbitrary point loads exerted by a human finger pressing on the edge of the beam. The beam was stimulated 8 times, each time with a non-constant intensity. Data from a trademark triaxial accelerometer BDI and a displacement transducer were recorded and saved on a PC. The BDI triaxial accelerometer has an accuracy of ±2 g. The adopted displacement transducer from KYOWA electronic instruments was of type DT-100A with a measuring capacity ranging between 0 and 100 mm. Both sensors were calibrated before the experiment.

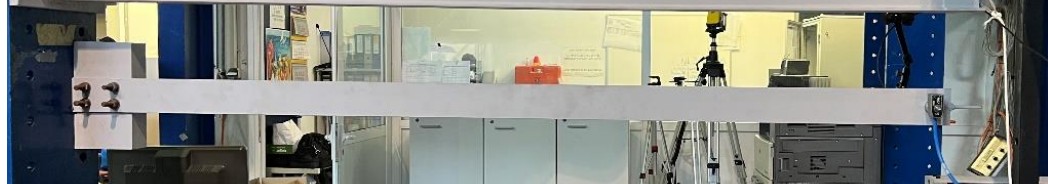

**Figure 4.** Cantilever beam fixed from one end and free from one end.

## 6. Developed LabVIEW Program Topology

### 6.1. Import the Acceleration–Time Domain Data into LabVIEW

First, the imported (.txt) file comprises the recorded acceleration data from the accelerometer in the X, Y, and Z directions and the displacement from the displacement transducer (Figure 5). The Z direction of the accelerometer, transverse to the beam, will be treated as the direction in which displacements were recorded by means of the displacement transducer. Figure 6 shows that the developed LabVIEW program is capable of counting the number of samplings and the duration of the recording time in second (s), the sampling frequency in (Hz), and the sampling rate.

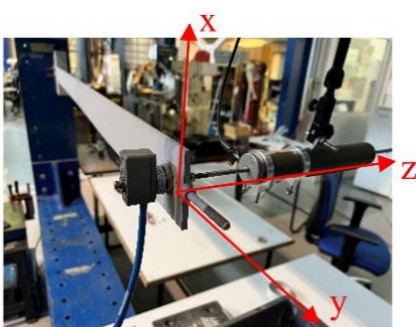

**Figure 5.** Displacement transducer and BDI accelerometer mounted on the cantilever beam tested under laboratory conditions.

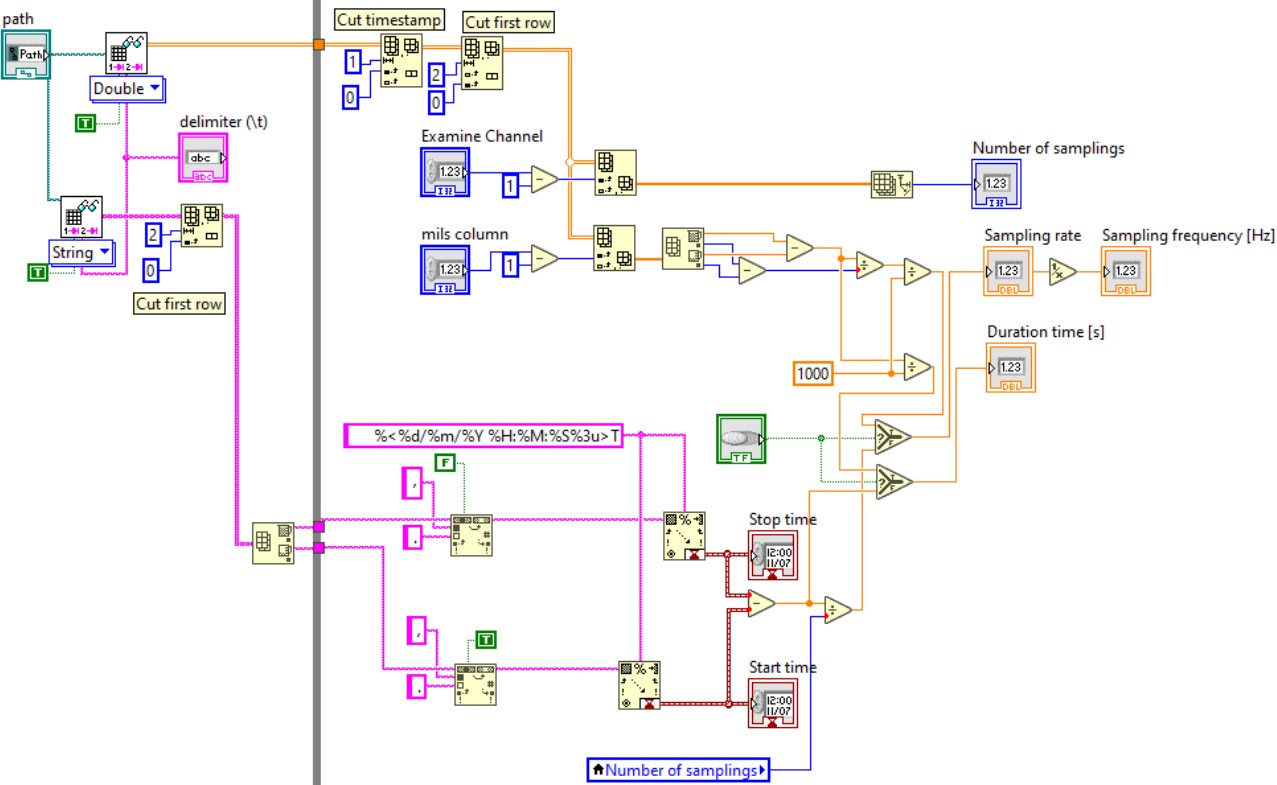

**Figure 6.** Section retrieved from the developed LabVIEW code showing the series of functions adopted to calculate the number of samplings, the sampling frequency, and the sampling rate.

*6.2. Calculate the Mean Value and the Corrected Acceleration–Time Domain*

The LabVIEW code calculates and eliminates the mean value from the recorded acceleration in order to remove the offset from the raw acceleration–time graph. This offset is considered a potential source of error if it is not eliminated. In order to calculate the corrected acceleration–time domain, the mean value is subtracted from the raw acceleration–time domain. Figure 7 shows that the calculated number of samplings and the sampling rate are used along with the LabVIEW function Amplitude and Level Measurements to calculate the mean value and subtract it from the recorded raw acceleration–time domain to finally estimate the corrected acceleration–time domain.

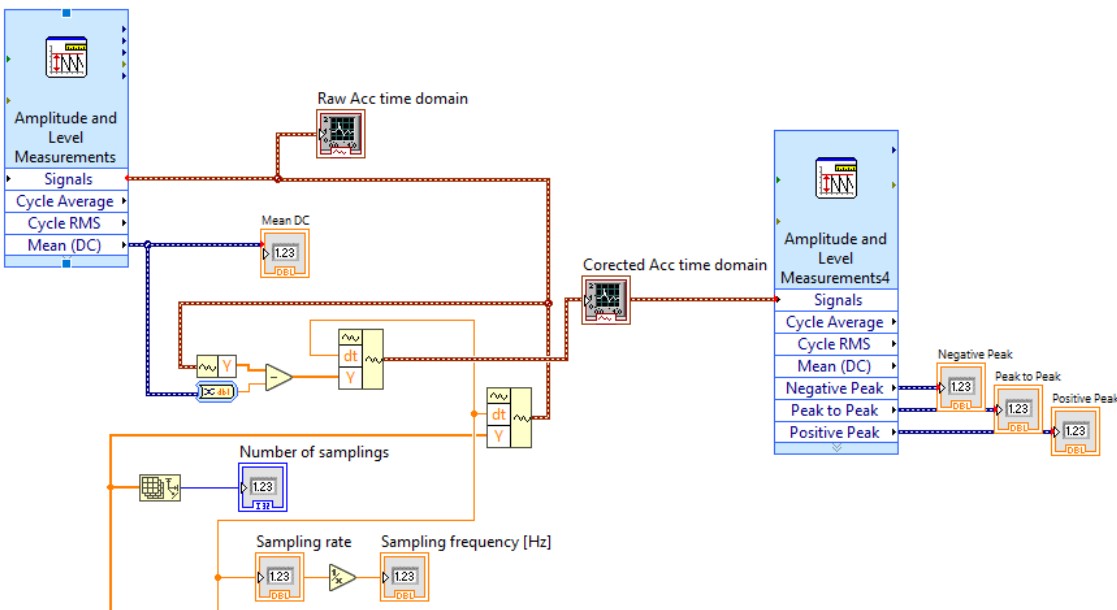

**Figure 7.** Section retrieved from the developed LabVIEW code showing the series of functions adopted to calculate the mean value and the corrected acceleration.

### 6.3. Calculate the System Eigenfrequencies

Figure 8 presents the corrected acceleration–time domain as it allows the user to select which direction (X, Y, or Z) to show on the graph. For the corrected acceleration–time domain, an FFT power spectrum and power spectral density (PSD) function was employed. The signal from the PSD was then evaluated to find the peaks in terms of the natural frequencies in (Hz) and their respective power in $\left[\frac{(\frac{m}{s})^2}{Hz}\right]$.

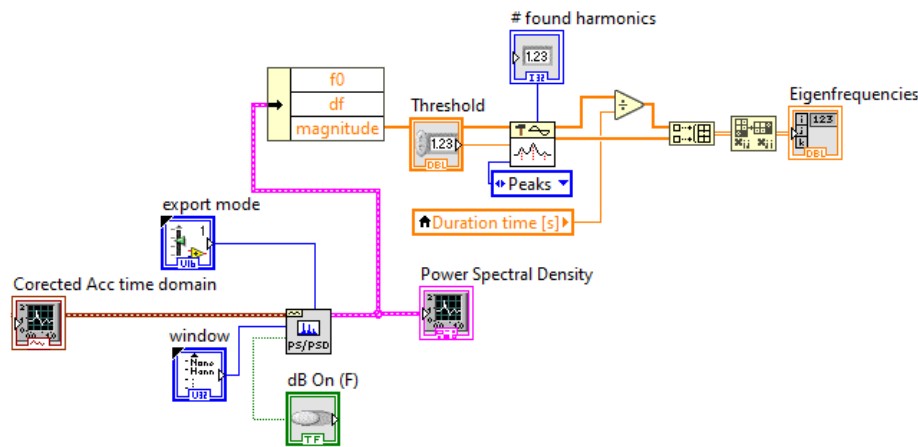

**Figure 8.** Section retrieved from the developed LabVIEW code showing the series of functions adopted to calculate the system eigenfrequencies.

### 6.4. Butterworth Filter

Figure 9 shows the application of a filter on the corrected acceleration–time domain. The LabVIEW function filter allows the user to select the filter design, the filter type, the order, and the cutoff frequencies.

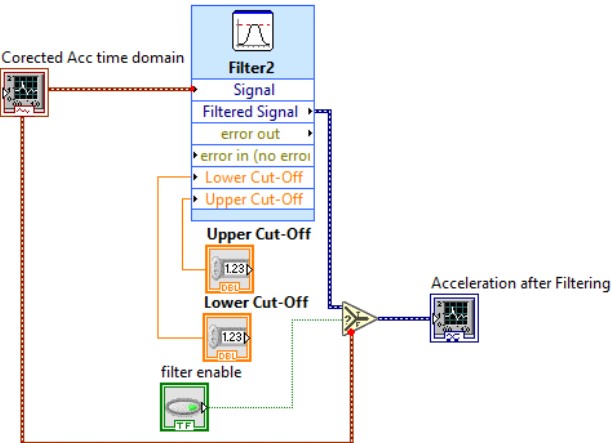

**Figure 9.** Section retrieved from the developed LabVIEW code showing the series of functions adopted to calculate and apply a filter to the corrected acceleration–time domain.

A second-order Butterworth filter was applied to the corrected acceleration–time domain with a bandpass design to calculate the damping ratio and a high pass to calculate the displacement, and setting appropriate frequency cutoffs.

### 6.5. Extraction of a Portion from the Filtered Acceleration

Figure 10 shows the adoption of the LabVIEW function Extract Portion of Signal that grants the user the preference of choosing a portion from the filtered acceleration–time domain by selecting its starting time and its duration.

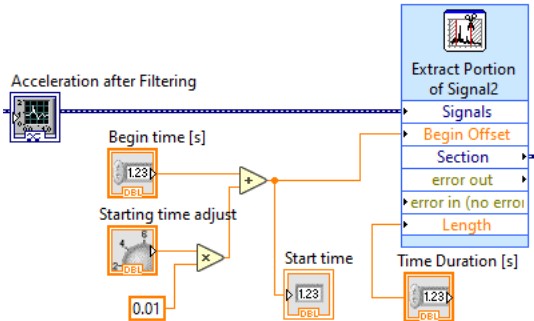

**Figure 10.** Section retrieved from the developed LabVIEW code showing the series of functions adopted to extract a portion from the filtered acceleration–time domain.

This portion will validate that the integrations start at zero acceleration and will undergo a series of mathematical operations in order to calculate the damping ratio and displacement.

### 6.6. Damping Ratio Calculation

For the calculation of the damping ratio, the Hilbert transform method along with the logarithmic decrement method were employed.

The Hilbert transform starts by selecting a portion of the corrected acceleration, where the segment of data should start at an amplitude of zero. A filter design, filter type, the order frequency, and the cutoff frequency are selected by the user. The damping can be calculated for every detected natural frequency. Then, a (rectangular) window is selected.

From the LabVIEW library, the Hilbert function was adopted to find the real and imaginary parts. From the real part, the upper envelop was formed and the damping ratio was retrieved from the exponential fit function, as it was divided by the natural frequency and multiplied by 100.

The same was applied to the imaginary part, which was multiplied by (−1), and then, the damping ratio was retrieved from the exponential fit function to be divided by the natural frequency and multiplied by 100 in order to turn it into a percentage (Figure 11).

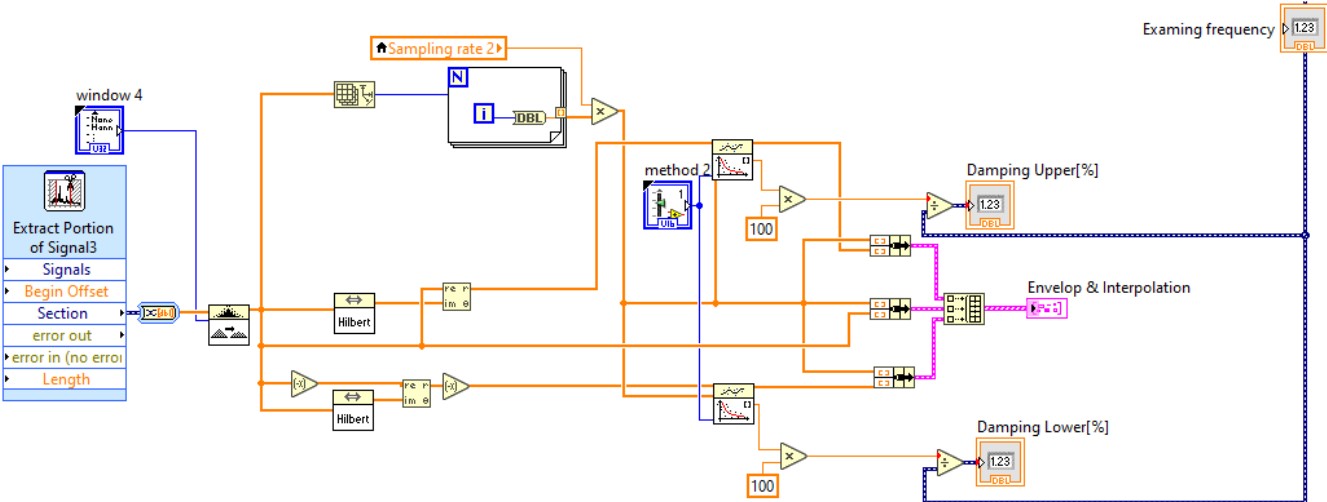

**Figure 11.** Sections retrieved from the developed LabVIEW code showing the series of functions adopted to calculate the damping ratio employing the Hilbert transform.

Similarly, the logarithmic decrement method requires the transformation of the acceleration signal into numerical values, with the selection of the filter design, filter type, the order frequency, and the cutoff frequency. Moreover, the selection of a (rectangular)window is required. Then, the selected acceleration–time domain is subjected to a peak detecting function, since damping will be calculated from the peaks, and then, the chosen acceleration peaks are turned into angular acceleration by being dividing by $[(2\pi)^2]$. Then, the exponential fit function is adopted in order to calculate the damping from the envelop equation.

The damping ratio is calculated after being divided by the natural frequency and multiplied by 100 in order to convert it into a percentage (Figure 12).

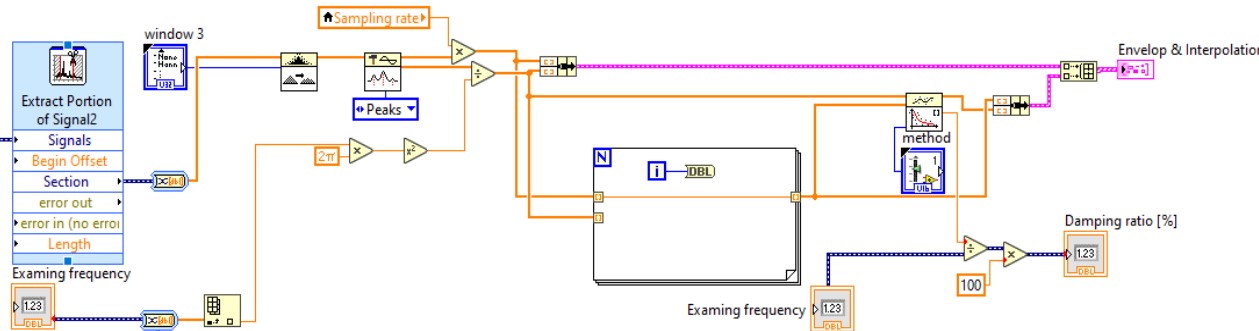

**Figure 12.** Section retrieved from the developed LabVIEW code showing the series of the function adopted to calculate the system damping ratio employing the logarithmic decrement method.

### 6.7. Displacement Calculation

For the displacement calculation, the corrected acceleration data are filtered and subjected to its first integration. Then, the mean value in the newly calculated velocity is evaluated to be eliminated so that no offset is present in the data. Then, the velocity is subjected to new integration so that the displacement data are calculated. In this program, displacement is presented in the unit of (mm) so it is multiplied by 1000, with its highest and lowest values (Figure 13).

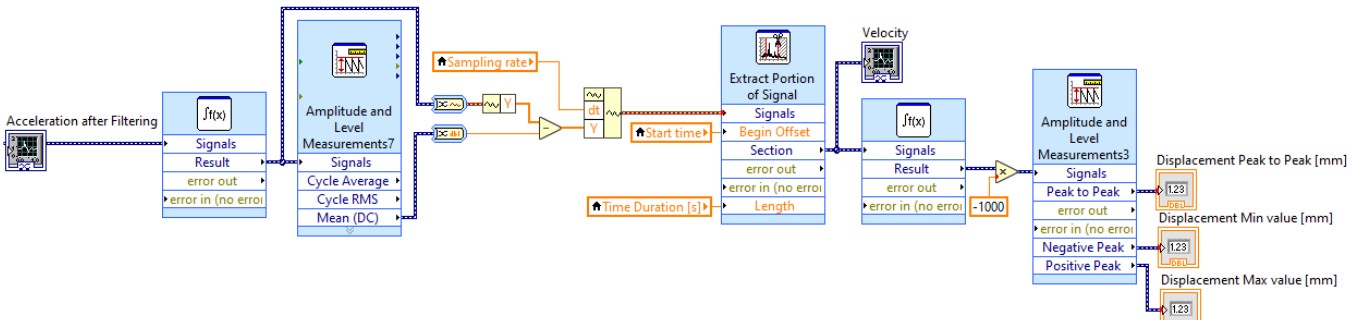

**Figure 13.** Section retrieved from the developed LabVIEW code showing the series of functions adopted to calculate the system displacement.

## 7. Estimation of Damping Parameters

Another important integral transform other than the Fourier transform is the Hilbert transform [72]. Identifying the damping properties is very significant in structural vibration. Modal parameters such as the damping of a structure might be calculated by computing the logarithmic decrement that occurs between two adjacent peaks of decreasing amplitudes in the time domain after a structure is stimulated. Unfortunately, the peaks are not precisely recorded in real experimental records. Due to either nonlinearity or noise included in the wave, a peak can a reach higher amplitude than the previous one leading to inaccurate decreasing damping [73]. Therefore, the Hilbert transform is implemented to effectively estimate the damping parameters.

Several researchers have suggested and implemented time–frequency assessment-based system techniques for parameter detection [74]. The Hilbert transform technique, like numerous other approaches, mainly the wavelet transform and empirical model decomposition [EMD], is involved in converting the recorded multifrequency response into one single frequency wave as a principal system response aiming to detect the system state and calculate the modal properties [75,76]. Xia et al., in 2021, developed an adaptive wave decomposition technique based on Hilbert vibration decomposition (HVD) for the purpose of acquiring the natural frequency and damping ratio. It is achieved by computing instantaneous system response for both linear and nonlinear systems. Whilst instantaneous natural frequency and the damping ratio are considered to be stable for a linear system, they have the tendency to fluctuate over time for nonlinear systems [5].

As stated by [73,77], the Hilbert transform of a signal $x(t)$ is the following:

$$H[x(t)] = \widetilde{x}(t) = \pi^{-1} \int_{-\infty}^{+\infty} \frac{x(\tau)}{t-\tau} d\tau \tag{1}$$

The integral is regarded to be a Cauchy principal value due to the potential singularity when $t - \tau = 0$.

The significance of the precedent equation allows the researcher to obtain a far more in-depth understanding of the transformation. HT is considered a specific linear filter in which the spectral components phases are moved by $\left[\frac{-\pi}{2}\right]$ without any magnitude modification [77].

Where $\widetilde{x}(t)$ is an imaginary signal of $-90°$ moved phase from the real signal $x(t)$,

$$X(t) = x(t) + \mathrm{j}\, \widetilde{x}(t) \tag{2}$$

The |vector sum| of the real and the imaginary signals is the amplitude:

$$|X|(t) = \sqrt{x^2(t) + \widetilde{x}^2(t)} \tag{3}$$

The amplitude of a decaying sinusoid is the envelope:

$$|X|(t) = A_{Oe^{-\zeta}}\omega_n t \tag{4}$$

Acceleration vs. time data were collected by accelerometers. After treating them with the corresponding filter and cleaning the wave from any unwanted noise, the acceleration vs. time graph has a cover equation to:

$$A(t) = A_{Oe^{-\zeta}}\omega_n t \tag{5}$$

where: $A_O$ = initial amplitude, $\zeta$ = damping ratio, and $\omega_n$ = natural frequency.

Results obtained in LabVIEW were compared with manual calculation for the natural frequency and the ARTeMIS modal analysis for both damping and displacement.

## 8. Benchmarking Information between LabVIEW and ARTeMIS

ARTeMIS and LabVIEW are both software programs commonly used for data acquisition and analysis in the field of vibrational testing. ARTeMIS is considered a reference code, as it was always implemented and tested by numerous sophisticated projects. However, the developed LabVIEW code makes a significant contribution to the field. Both software differ in their approach to uploading data.

Both programs demand the upload of the numerical data recorded by the accelerometers mounted on the structure, along with the specification of the sensor's sampling rate before running the codes. It is noted that LabVIEW will not require the user to draw the geometry of the structure or to specify the location of each sensor mounted on the structure. Whereas ARTeMIS will request the drawing of the structure, the specification of the locus of each sensor and allocation of each uploaded data with the specific directory (x,y,z) as the adopted accelerometers are triaxial.

The simplicity of the input required for LabVIEW will protect the user from committing any error and will save time. Moreover, it does not require one to readjust the code when working on different projects. Each time, the user will upload the measurements recorded by the sensors and will only specify the sampling rate.

Once the code runs, LabVIEW will provide the calculated measurements from natural frequency, damping, and displacement for each accelerometer, whereas ARTeMIS will show the response given by the most amplified signal.

For comparison reasons, the modal parameters were calculated with the ARTeMIS modal analysis software using enhanced frequency domain decomposition (EFDD) methodology. It is important to identify the deterministic signals and to minimize their contribution when calculating the vibrational measurements of the system after OMA due to the fact that the employed force is not predetermined.

ARTeMIS presents the various decomposition methodologies that a researcher can adopt to conduct an OMA. Among these, there are the frequency domain decomposition (FDD) and the enhanced frequency domain decomposition (EFDD) [78]. FDD is considered an effortlessly applicable decomposition method. However, it does not lead to damping calculation, and it revolves around FFT analysis to effectively assess the natural frequency of a system as it only uses an individual frequency channel. An addition to the FDD approach is the EFDD technique, which provides a better estimation of the eigenfrequency and takes damping calculation into account. Employing the EFDD enables one to assess both the natural frequency and the damping ratio with great precision. Both the natural frequencies and damping ratio were assessed in ARTeMIS for the sake of comparing the results conducted on the newly developed LabVIEW program.

The developed LabVIEW program mainly relies on mathematical functions such as fast Fourier transform (FFT) or power spectral density (PSD) for natural frequency calculation, and Hilbert transform (HT) as well as the logarithmic decrement method for damping calculation. These methods are considered the most suitable techniques and have been adopted in many projects and research papers. For instance, FFT was adopted by Lin and

Yang in 2005 for estimating the natural frequency of the Da-Wu-Lun bridge located in Taiwan [79]. In 2007, Mengh et al. employed the FFT technique to calculate the natural frequency of the pedestrian Wilford Bridge located over River Trent in Nottingham [80]. By employing the hamming spectral window operator, FFT enables the determination of a power spectral density. The same was performed for the damping calculation, where, on the one hand, this research study adopted the HT, as it was the most adopted by other researchers. In 2014, Gonzalez and Karoumi calculated the damping of a bridge situated on the northern part of Sweden using HT [81]. In 2004, Chen et al. adopted a newly emerged empirical mode decomposition (EMD) technique along with HT (EMD-HT) to calculate the damping of the Tsing Ma suspension bridge located in Hong Kong [82]. On the other hand, a logarithmic decrement was also adopted, as Nakutis and Kaskonas calculated the damping ratio of a two-girder bridge in 2010 using the logarithmic decrement approach [83]. In 2020, Chen et al. estimated the damping of long cables with the viscous-shear dampers of the Sutong cable-stayed bridge located in China [84].

## 9. Results

The data of 44,565 samplings collected with a sampling frequency of 500 Hz and a sampling rate of 0.002 were saved in a text file. These are the data collected by both the accelerometer and the displacement transducer. Both are presented in the time domain (Figures 14 and 15).

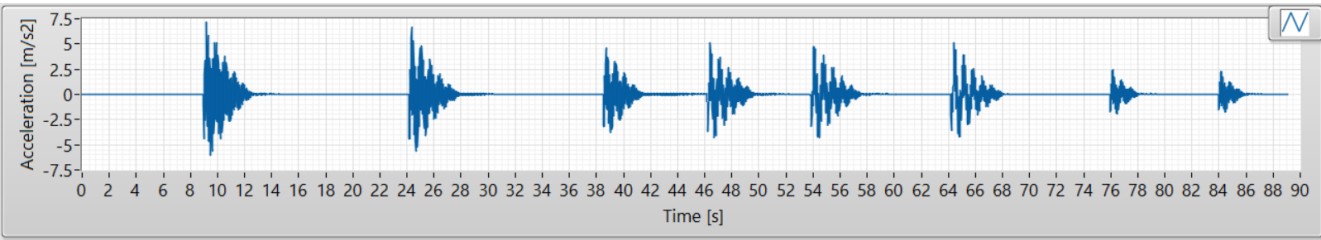

**Figure 14.** LabVIEW graph showing the corrected acceleration–time domain recorded by the accelerometer mounted on the cantilever beam.

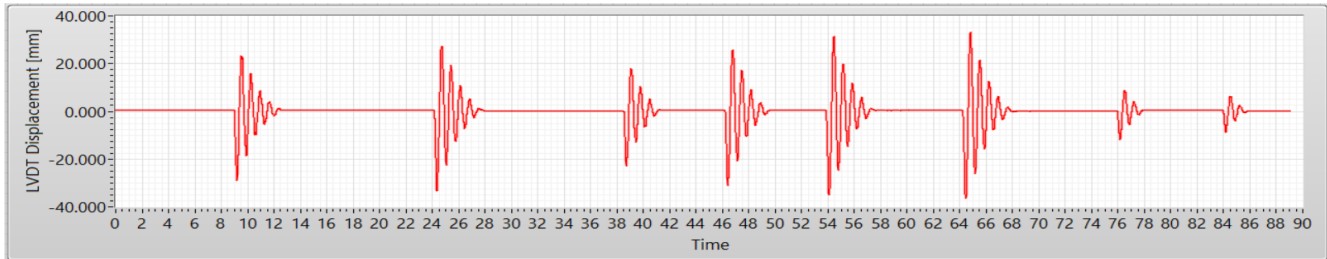

**Figure 15.** LabVIEW graph showing the displacement–time domain recorded by the displacement transducer mounted on the cantilever beam.

### 9.1. Natural Frequency

For steel: the Young's modulus E = $2 \times 10^{11}$ N/m$^2$
The unit weight ρ = 7850 kg/m$^3$
b = 0.08 m, h = 0.006 m, L = 1.86 m
The area: A = b × h = 0.00048 m$^2$
The moment of inertia: $I = \frac{1}{12} b\, h^3 = 1.44 \times 10^{-9}$ m$^4$

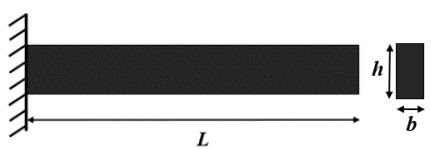

For the cantilever beam, the angular frequency $\omega_n = \sqrt{\frac{E\,I}{\rho\,A}\frac{(\alpha_n)^2}{L^2}}$, for $n$ (number of modes) = 1, 2, 3.

Where $\alpha_1 = 1.875$, $\alpha_2 = 4.694$, $\alpha_3 = 7.855$ according to [85].
Therefore, $\omega_1 = 8.88$ rad/s, $\omega_2 = 55.68$ rad/s, $\omega_3 = 155.92$ rad/s.

The natural frequency $f_n = \frac{\omega_n}{2\pi}$, therefore $f_1$ = 1.413 Hz, $f_2$ = 8.861 Hz, $f_3$ = 24.815 Hz.

LabVIEW was capable of extracting the first two natural frequencies equivalent to 1.4456 Hz and 8.217 Hz, with an error of 2.2% and 7.2%, respectively (Figures 16 and 17).

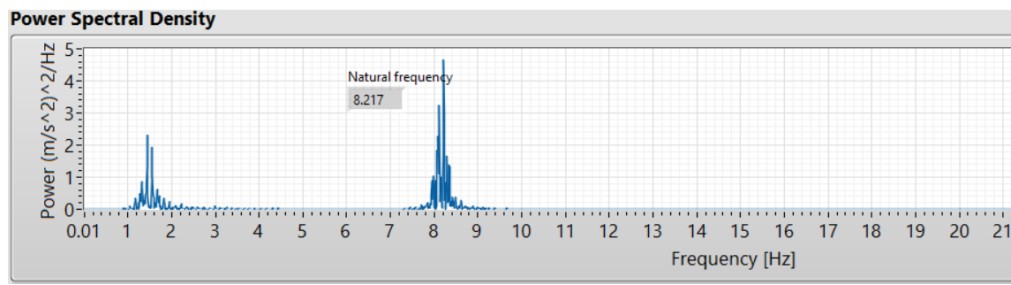

**Figure 16.** LabVIEW graph showing the power spectral density with a numerical inquiry of the system eigenfrequency and its respective amplitude.

**Eigenfrequencies**

| | |
|---|---|
| 1.4456 | 2.36165 |
| 8.21444 | 4.68952 |
| 0 | 0 |

**Figure 17.** LabVIEW table showing the power spectral density graph with numerical inquiry of the system eigenfrequency and its respective amplitude.

For ARTeMIS, on the other hand, the first two natural frequencies were 1.396 Hz and 8.148 Hz, with errors of 3.39% and 0.84%, respectively, with that of the LabVIEW (Figures 18 and 19).

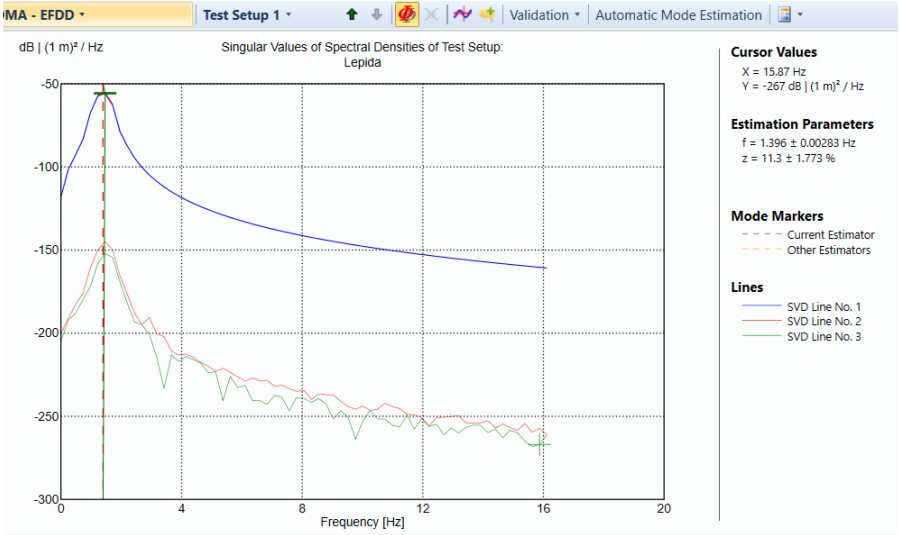

**Figure 18.** ARTeMIS modal analysis operational modal analysis using the EFDD technique and showing the calculated first eigenfrequencies.

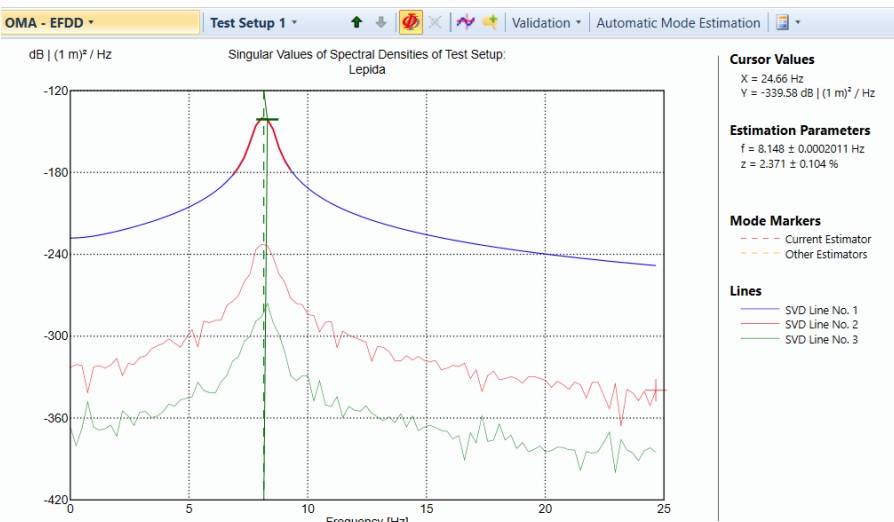

**Figure 19.** ARTeMIS modal analysis operational modal analysis using the EFDD technique and showing the calculated second eigenfrequencies.

*9.2. Damping*

LabVIEW was programmed to calculate the damping ratio through the Hilbert transform and logarithmic decrement method (Figure 20).

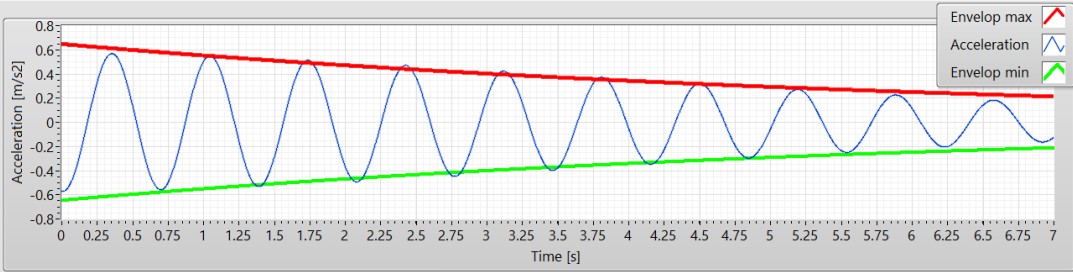

**Figure 20.** LabVIEW graph showing the acceleration–time domain with envelops for the first eigenfrequency calculated by the Hilbert transform.

The damping ratio was calculated around the two found natural frequencies of the beam. For the first eigenfrequency, a second-order Butterworth bandpass filter was applied on the acceleration–time domain recorded data, with cutoff frequencies between 1.40 and 1.49 Hz. According to the Hilbert transform, the damping ratio is 11% (Figure 21).

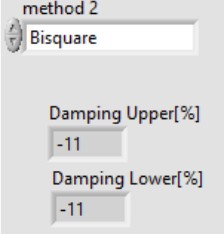

**Figure 21.** LabVIEW numerical analysis showing the damping ratio for the first eigenfrequency.

According to the Logarithmic decrement method, the damping recorded for the first eigenfrequency was 11% (Figure 22).

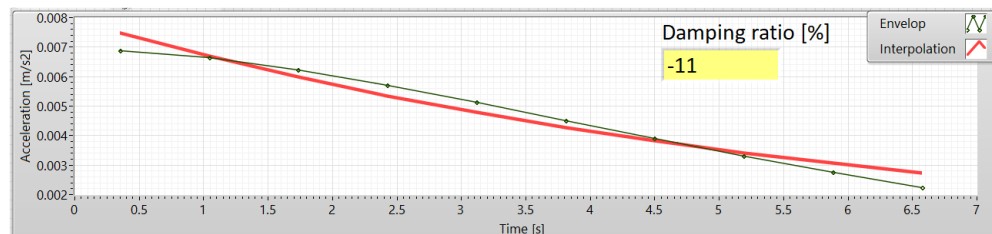

**Figure 22.** LabVIEW graph showing the logarithmic decrement calculation of a damping ratio for the first eigenfrequency.

The error margin between Hilbert transforms and the logarithmic decrement is 0% for the first eigenfrequency. For the second eigenfrequency, a second-order Butterworth bandpass filter was applied on the acceleration–time recorded data, with cutoff frequencies between 8.1 and 8.2 Hz (Figure 23).

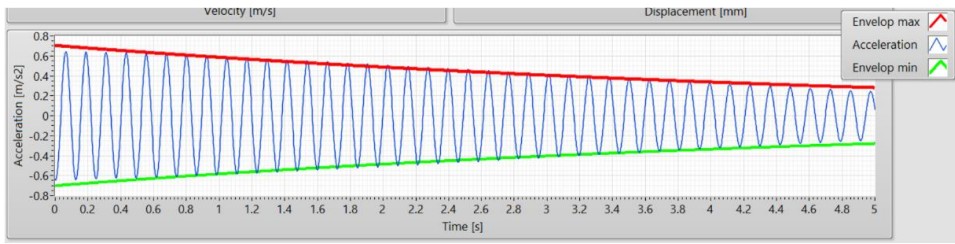

**Figure 23.** LabVIEW graph showing the acceleration–time domain with envelops for the second eigenfrequency calculated by the Hilbert transform.

According to the Hilbert transform, the damping ratio is 2.26% (Figure 24).

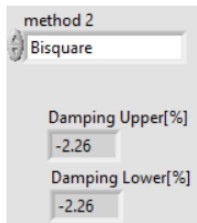

**Figure 24.** LabVIEW numerical analysis showing the damping ratio for the second eigenfrequency.

According to the logarithmic decrement method, the damping recorded for the first eigenfrequency was 2.3% (Figure 25).

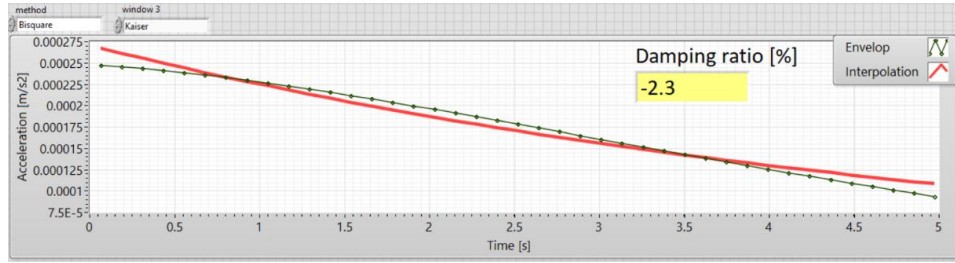

**Figure 25.** LabVIEW graph showing the logarithmic decrement calculation of the damping ratio for the second eigenfrequency.

The error margin between Hilbert transforms and the logarithmic decrement is 1.7% for the second eigenfrequency. The same analysis was conducted in the ARTeMIS modal standard, and for the first eigenfrequency, a second-order Butterworth bandpass filter was

applied on the acceleration–time recorded data, with cutoff frequencies between 1.40 and 1.49 Hz. OMA was conducted using the EFDD technique [86] to estimate the damping around the first eigenfrequency.

According to OMA using the EFDD methodology in ARTeMIS, the damping recorded for the first eigenfrequency was 11.295% with an error of 2.61% between ARTeMIS and both the Hilbert transform and logarithmic decrement conducted in LabVIEW (Figure 26).

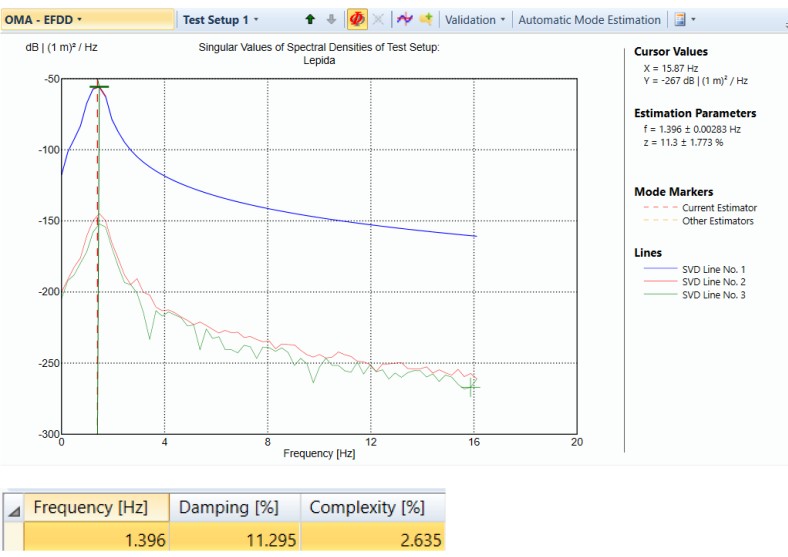

**Figure 26.** ARTeMIS graph OMA through EFDD revealing the damping for the first eigenfrequency.

The same analysis was conducted in the ARTeMIS modal standard, and for the second eigenfrequency, a second-order Butterworth bandpass filter was applied on the acceleration–time recorded data, with cutoff frequencies between 8.1 and 8.2 Hz. According to OMA using the EFDD methodology in ARTeMIS, the damping recorded for the second eigenfrequency was 2.371% with an error of 4.68% between the ARTeMIS and Hilbert transform conducted in LabVIEW, and 2.99% between ARTeMIS and the logarithmic decrement conducted in LabVIEW (Figure 27).

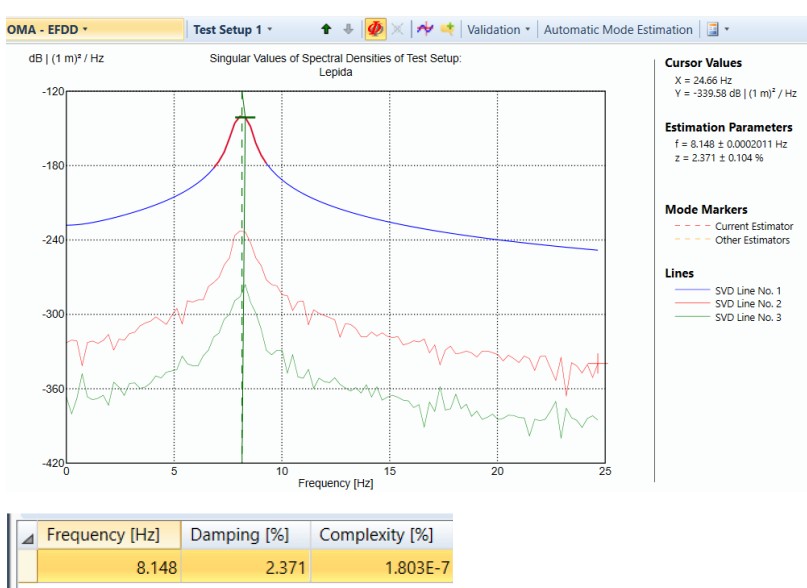

**Figure 27.** ARTeMIS graph OMA through EFDD revealing the damping for the second eigenfrequency.

### 9.3. Displacement

The displacement transducer was mounted on the structure for comparison purposes. It recorded the displacement versus time for the eight excitations that the structure was subjected to. It recorded the highest displacement at the sixth excitation with a peak of 69.9 mm (Figures 28 and 29).

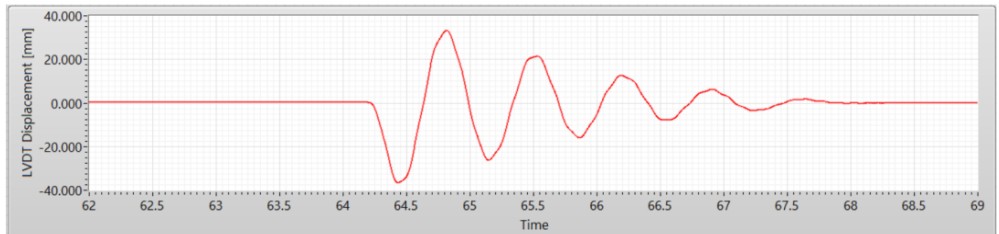

**Figure 28.** LabVIEW graph showing displacement transducer displacement–time domain.

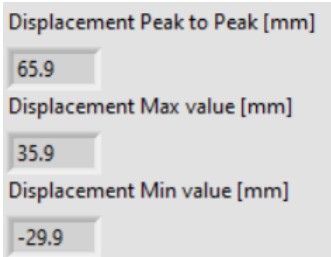

**Figure 29.** LabVIEW numerical analysis showing the displacement recorded by displacement transducer.

The acceleration–time data retrieved from the mounted accelerometer were treated be doubly integrated and enable the calculation of the displacement (Figure 30).

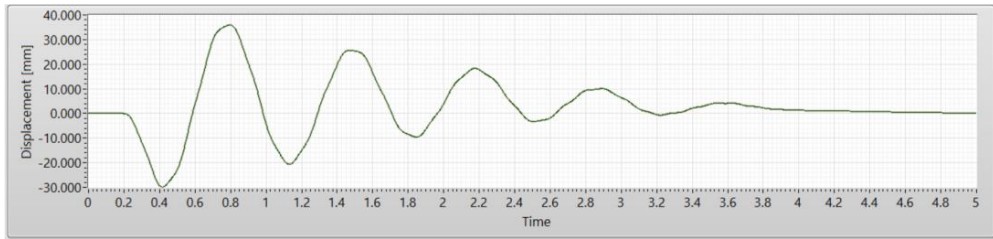

**Figure 30.** LabVIEW graph showing the displacement–time domain calculated from the accelerometer.

A second-order high-pass Butterworth filter was applied on the raw acceleration data with a low cutoff frequency of 0.2 Hz. It recorded the highest displacement at the same peak as the displacement transducer with a displacement equivalent to 65.9 mm. An error of 5.7% between the displacement transducer and the displacement was retrieved from the accelerometer (Figure 31).

**Displacement Peak to Peak [mm]**
65.9
**Displacement Max value [mm]**
35.9
**Displacement Min value [mm]**
-29.9

**Figure 31.** LabVIEW numerical analysis showing the displacement calculated from the acceleration data.

The same data were treated in ARTeMIS which showed that the displacement transducer was capable of recording the same highest displacement value at the sixth peak.

For acceleration–time data, ARTeMIS is capable with of doubly integrating these with a high-pass Butterworth filter with a low cutoff frequency equivalent to 0.2 Hz, but only with order 1.

The displacement calculated from the acceleration–time data in ARTeMIS is 62.7 mm (Figure 32). With an error of 10.35%, the ARTeMIS is capable of double integration using filtered acceleration data of the first order. According to the results concerning the natural frequency, damping, and displacement calculation, LabVIEW was capable of calculating all these parameters with an error of less than 5%, as it outperformed ARTeMIS as well as gave the researcher the capability and the flexibility to choose a better filter, leading to better results.

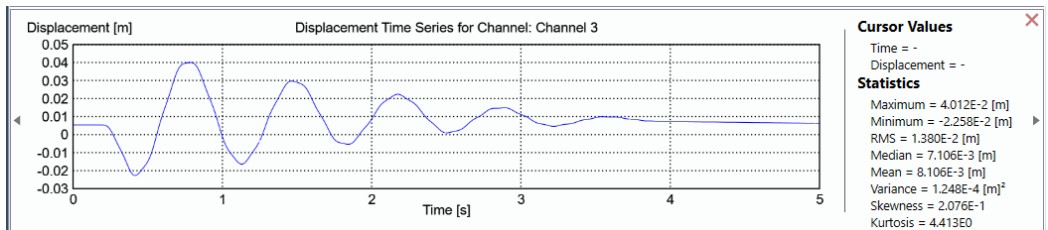

**Figure 32.** ARTeMIS graph showing displacement–time domain calculated from the recorded acceleration data.

## 10. Conclusions

In this study, a LabVIEW program was developed for vibrational monitoring and system evaluation. It comprises a data acquisition and evaluation technique. The established code is capable of reading the data collected by an accelerometer mounted on a structure in the time domain, apply FFT and PSD in order to shift to the frequency domain, and extract the eigenfrequencies from the highest recorded peaks. The collected measurements were filtered for damping and displacement calculations. A Butterworth filter was adopted with bandpass cutoff frequencies to surround each calculated natural frequency and calculate the damping ratio through employing both techniques, namely the Hilbert transform and the logarithmic decrement. The same filter was adopted with a high-pass cutoff frequency to account for the low frequency noise coming either from the accelerometer itself or from the background in order to doubly integrate the acceleration and calculate the displacement. As a consequence of building-in all the aforementioned functions, the identification of modal parameters such as eigenfrequencies, damping ratios, and displacement has become effortlessly accessible. As it overcomes numerous challenges that arise in vibrational measurement, identification allows its operation even by unexperienced users.

The developed program was capable of detecting the eigenfrequencies, damping, and displacement from the acceleration data. The evaluated parameters were estimated with the ARTeMIS modal analysis software for comparison purposes. Additionally, manual calculation was performed to calculate the eigenfrequencies. Additionally, the displacement from the displacement transducer was accounted for in the displacement comparison.

The calculated natural frequencies were 1.4456 and 8.217 Hz. A second-order Butterworth bandpass filter was adopted for the damping ratio evaluation. For the first natural frequency, the cutoff frequencies were 1.40 and 1.49 Hz, and the damping ratio was 11% for Hilbert and the logarithmic decrement. For the second natural frequency, the cutoff frequencies were 8.1 and 8.2 Hz, and the damping ratio was 2.26 and 2.3%, respectively. A second-order Butterworth high-pass filter was adopted for the displacement calculation with cutoff frequency equal to 0.2 Hz. For the displacement, the double integration of the acceleration data showed a displacement of 65.9 mm, as shown in Table 1 below.

**Table 1.** A summary of the results obtained with the ARTeMIS modal analysis and the developed LabVIEW program and through manual calculation.

| Parameters | Manual Calculation | LabVIEW | ARTeMIS |
|---|---|---|---|
| Natural frequency | 1.41 Hz and 8.86 Hz | 1.44 Hz and 8.21 Hz | 1.39 Hz and 8.14 Hz |
| Damping | | 11% and 2.3% | 11.29% and 2.37% |
| Displacement | 69.9 mm | 65.9 mm | 62.7 mm |

With an error of less than 5%, LabVIEW was capable of accurately calculating the vibrational properties and the displacement. It outperformed the ARTeMIS software in the available filtering order in the displacement calculation.

The reported response confirmed that the proposed system strongly conducted the desired performance as it successfully identified the system state and the modal parameters.

**Author Contributions:** Conceptualization, R.E.D.; Methodology, R.E.D. and I.V.; Software, R.E.D., X.L. and S.P.; Validation, R.E.D., X.L and I.V.; Formal analysis, R.E.D.; Writing—original draft, R.E.D.; Writing—review & editing, I.V.; Supervision, I.V. All authors have read and agreed to the published version of the manuscript.

**Funding:** This research received no external funding.

**Data Availability Statement:** Some or all data, models, or code that support the findings of this study are available from the authors upon reasonable request.

**Conflicts of Interest:** The authors declare no conflict of interest.

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
