# Peer review of "Development and Validation of a LabVIEW Automated Software System for Displacement and Dynamic Modal Parameters Analysis Purposes"

_2673-3951, doi:10.3390/modelling4020011_

Round 1

Reviewer 1 Report

Modal testing is a well-studied area. The content presented in the manuscript is a commonly applied approach. It is difficult for the reviewer to grasp any novelties and/or significant contributions to the area. 

Author Response

Response to Reviewer 1 Comments

We would like to thank you for taking the time to review our manuscript. We appreciate your feedback and understand your concern regarding the novelty of our developed LabVIEW code.

While we acknowledge that modal testing is a well-studied area, we believe researchers can still find gaps when it comes to post processing.

ARTeMIS and LabVIEW are both software programs commonly used for data acquisition and analysis in the field of vibrational testing.

ARTeMIS is considered a reference code, as it was always implemented and tested by numerous sophisticated projects. However, we do think that the developed LabVIEW code still makes a significant contribution to the field. Both softwares differ in their approach to uploading data.

For that reason, we have established a comparison between them. In our revised manuscript, we will provide a more detailed discussion on benchmarking information regarding the uploaded data to both programs prior to post processing estimation and measurements calculation, in addition after running, the accuracy of the adopted mathematical functions was checked:

“Both programs demand the upload of the numerical data recorded by the accelerometers mounted on the structure, along with specifying the sensor’s sampling rate before running the codes.

It is noted that LabVIEW will not require from the user to draw the geometry of the structure neither to specify the location of each sensor mounted on the structure. Whereas ARTeMIS will request the drawing of the structure, the specification of the locus of each sensor and the allocation of each uploaded data with the specific directory (x,y,z) as the adopted accelerometers are triaxial.

The simplicity of the input required for LabVIEW will protect the user from committing any error and will save time. Moreover, it does not require to readjust the code when working on different projects. Each time, the user will upload the recorded measurements by the sensors and will only specify the sampling rate.

Once the code runs, LabVIEW will provide the calculated measurements from natural frequency, damping and displacement for each accelerometer, whereas ARTeMIS will show the response given by the most amplified signal.

The developed LabVIEW program relies mainly on mathematical functions such as Fast Fourier Transform (FFT) or Power Spectral Density (PSD) for natural frequency calculation, and Hilbert Transform (HT) as well as logarithmic decrement method for damping calculation. As these methods are considered the most suitable techniques since they were adopted in many projects and research papers.

For instance, FFT was adopted by Lin & Yang, in 2005 for estimating the natural frequency of the Da-Wu-Lun bridge located in Taiwan [1]. Mengh et al., in 2007 have employed FFT technique to calculate the natural frequency of the pedestrian Wilford Bridge located over River Trent in Nottingham [2].

By employing the hamming spectral window operator, FFT enables the determination of a power spectral density.

Same for the damping calculation, on one hand this research study has adopted the HT, as it was the most employed by other researchers. Gonzalez & Karoumi in 2014 have calculated the damping of a bridge situated on the northern part of Sweden using HT [3]. Chen et al., in 2004 adopted a new emerged empirical mode decomposition (EMD) technique along with HT (EMD-HT) to calculate the damping of Tsing Ma suspension bridge located in Hong Kong [4].

On the other hand, logarithmic decrement was employed as well, Nakutis & Kaskonas in 2010 have calculated the damping ratio of a two-girder bridge using logarithmic decrement approach [5]. Chen et al. in 2020 have estimated the damping of long cables with viscous-shear dampers of the Sutong cable-stayed bridge located in China [6].

After profiting from all these mathematical functions in the developed LabVIEW code and treating the case of the beam in the steel cantiliver beam on both LabVIEW and ARTeMIS for comparison purposes, the results showed a difference less than 5% which proves the accuracy and efficiency of LabVIEW. ”

[1] Lin, C. W., & Yang, Y. B. Use of a passing vehicle to scan the fundamental bridge frequencies: An experimental verification. Engineering Structures 2005, 27(13), pp.1865-1878.

[2] Meng, X., Dodson, A. H., & Roberts, G. W. Detecting bridge dynamics with GPS and triaxial accelerometers. Engineering Structures 2007, 29(11), pp.3178-3184.

[3] Gonzalez, I., & Karoumi, R. Analysis of the annual variations in the dynamic behavior of a ballasted railway bridge using Hilbert transform. Engineering structures 2014, 60, pp.126-132.

[4] Chen, J., Xu, Y. L., & Zhang, R. C. Modal parameter identification of Tsing Ma suspension bridge under Typhoon Victor: EMD-HT method. Journal of Wind Engineering and Industrial Aerodynamics 2004, 92(10), pp.805-827.

[5] Nakutis, Ž., & Kaškonas, P. Bridge vibration logarithmic decrement estimation at the presence of amplitude beat. Measurement 2011, 44(2), pp.487-492.

[6] Chen, L., Di, F., Xu, Y., Sun, L., Xu, Y., & Wang, L. Multimode cable vibration control using a viscous‐shear damper: Case studies on the Sutong Bridge. Structural Control and Health Monitoring 2020, 27(6), p. e2536.

We hope that these contributions will be of interest to the readers of this journal and help advance the field of modal testing and post processing. Once again, thank you for your feedback, and we look forward to addressing any further comments or concerns you may have.

Point 1: Does the introduction provide sufficient background and include all relevant references?

Response 1: We have provided some background information on structural health monitoring (SHM) and its importance in assessing and maintaining the structural integrity of civil engineering infrastructure. It also briefly explains the use of vibrational parameters and displacement calculation in SHM and mentions operational modal analysis (OMA) as a method for identifying changes in structural parameters.

As our main concern was introducing the developed LabVIEW coding technique for calculating vibrational parameters, such as natural frequencies and damping, and displacement measurements.

And how it has contributed in decreasing the computational burden, the ease of implementation, and the accuracy of the results was considered. We have discussed in succeeding paragraphs about the importance of calculating these measurements and how they influence on the structural assessment and how to obtain each of these discussed measurements.

Even though the introduction had established a brief presentation on all the aformentioned aspects, but through the research structure we believe we had covered all needed topics for the reader to understand the aim of the study.  

Point 2: Are all the cited references relevant to the research?

Response 2: We made sure that every reference was relevant to one of the scientific aspects discussed in this research and it is mandatory to site them just to avoid plagiarism or not give credits to previous scientific work.

Point 3: Is the research design appropriate?

Response 3: Indeed, we have improved the research design in the revised manuscript as we added the description of the developed code benchmark and compare it with ARTeMIS, moreover , we have provided a more detailed discussion on benchmarking information regarding the uploaded data to both programs prior to post processing estimation and measurements calculation, in addition after running, the accuracy of the adopted mathematical functions was checked. Therefore we have covered any confusion the reader can get through to understand how each of the programs operates in terms of mathematical functions to calculate the measurements, and choose what is best for him according to the ease of implementation and the computational burden.

Furthermore, we have added how Numerous source of error were eliminated by developing the LabVIEW code. As they can be divided into two categories. First, the code itself is capable to eliminate them without the interference of the user such as the case of deleting the DC value from the raw acceleration-time data recorded by the accelerometers.

Therefore we have added this sentence in line 223:

“Figure 6 shows that the LabVIEW code calculates and eliminates the Mean (DC) value from the recorded acceleration in order to remove the offset from the raw acceleration time graph. This offset is considered as a potential source of error if it not eliminated.” 

Second, the developed LabView has saved the user by not asking for the following input data before running the code.

Among them we can site drawing the geometry incorrectly or forgetting to add a structural element especially if the studied structure is considered complicated. Not forgetting the error that can happen if the user makes a mistake in incorrectly specifying the locus of each sensor mounted on the structure.

We have made sure to include these comments in the same paragraph that we have added for the sake for the first comment you have mentioned.

Point 4: Are the methods adequately described?

Response 4: Undoubtedly, the method description can be improved. Even though we have taken some steps to adequately describe the methods used in the research study, particularly in terms of explaining how to implement the mathematical functions from the LabVIEW library to calculate and analyse each of the vibrational measurements and displacement Moreover, we made sure to describe in each section entitled after a specific measurement how they are calculated in a scientific way. And how we have implemented the scientific meaning into LabVIEW coding language.

However, we also acknowledge that the method description can be improved, and we plan to address this in the revised manuscript by providing some benchmarking information between the LabVIEW Vi on the one hand and the ARTeMIS software on the other hand and adding information on how the code functions overall, in terms of execution times and their dependency on sampling rates, and what data is needed to implement so it can run successfully.

As the performance of the developed code is discussed where the data acquisition and processing using Fast Fourier Transform (FFT) or Power Spectral Density (PSD) for natural frequency calculation, and Hilbert Transform (HT) as well as logarithmic decrement method for damping calculation. As we proved that adopting these techniques can easily convert to the optimised results as LabVIEW was able to outperform compiled source code using optimised mathematical libraries in terms of uploaded data and time, with same accuracy of the results. 

Point 5: Are the results clearly presented?

Response 5: In our research study we have taken appropriate measures to ensure that the results are clearly presented and well-supported. We made sure to present the results in figures directly retrieved from the softwares to ensure the authenticity of the outcome. Additionally, we compared the results with manual calculation for the case of the natural frequency calculation, and in the case of displacement we have compared the result of both softwares with a real time displacement transducer, which adds to the credibility of the findings. Finally, we have shown the difference among the calculated results in terms of percentage error which helps to quantify the accuracy of the results.

Reviewer 2 Report

While your paper on the development and validation of a LabVIEW automated software system for displacement and dynamic modal parameters analysis purposes presents an interesting approach to automated analysis, there are a few areas where it could be improved. For example, the paper could have provided more detailed information on the implementation of the system and how it compares to existing methods. Additionally, while the validation of the system's accuracy is commendable, the paper could have included more discussion on potential sources of error and how they were addressed. Furthermore, it would be beneficial if the paper could have included a more comprehensive evaluation of the system's performance under different conditions. Overall, while the paper presents a promising approach to automated analysis, more detailed information and evaluation are needed to fully assess its potential impact in the filed of structural health monitoring.

Author Response

Response to Reviewer 2 Comments

We would like to thank you for your valuable remarks that we took into consideration for fulfilling the needed criteria in order to publish our research study in your reputable journal.

Point 1: The paper could have provided more detailed information on the implementation of the system and how it compares to existing methods.

Response 1: ARTeMIS and LabVIEW are both softwares programs commonly used for data acquisition and analysis in the field of vibrational testing.

ARTeMIS is considered a reference code, as it was always implemented and tested by numerous sophisticated projects. For instance, the ambient assessment of the Vasco da Gama bridge located in Portugal was analysed on ARTeMIS during the design validation stage. Moreover, it was adopted in monitoring smaller bridge projects, the S101 highway bridge of Austria, in order to assess the damage and degradation over time bought by environmental influences [1].

However, both softwares differ in their approach to uploading data.

For that reason, we have established a comparison between them. We have added benchmarking information regarding the uploaded data to both programs prior to post processing estimation and measurements calculation, in addition after running, the accuracy of the adopted mathematical functions was checked:

“Both programs demand the upload of the numerical data recorded by the accelerometers mounted on the structure, along with specifying the sensor’s sampling rate before running the codes.

It is noted that LabVIEW will not require from the user to draw the geometry of the structure neither to specify the location of each sensor mounted on the structure. Whereas ARTeMIS will request the drawing of the structure, the specification of the locus of each sensor and the allocation of each uploaded data with the specific directory (x,y,z) as the adopted accelerometers are triaxial.

The simplicity of the input required for LabVIEW will protect the user from committing any error and will save time. Moreover, it does not require to readjust the code when working on different projects. The user will upload the recorded measurements by the sensors and will only specify the sampling rate.

Once the code runs, LabVIEW will provide the calculated measurements from natural frequency, damping and displacement for each accelerometer, whereas ARTeMIS will show the response given by the most amplified signal.

The developed LabVIEW program relies mainly on mathematical functions such as Fast Fourier Transform (FFT) or Power Spectral Density (PSD) for natural frequency calculation, and Hilbert Transform (HT) as well as logarithmic decrement method for damping calculation. As these methods are considered the most suitable techniques since they were adopted in many projects and research papers.

For instance, FFT was adopted by Lin & Yang, in 2005 for estimating the natural frequency of the Da-Wu-Lun bridge located in Taiwan [2]. Mengh et al., in 2007 have employed FFT technique to calculate the natural frequency of the pedestrian Wilford Bridge located over River Trent in Nottingham [3].

By employing the hamming spectral window operator, FFT enables the determination of a power spectral density.

Same for the damping calculation, on one hand this research study has adopted the HT, as it was the most adopted by other researchers. Gonzalez & Karoumi in 2014 have calculated the damping of a bridge situated on the northern part of Sweden using HT [4]. Chen et al., in 2004 adopted a new emerged empirical mode decomposition (EMD) technique along with HT (EMD-HT) to calculate the damping of Tsing Ma suspension bridge located in Hong Kong [5].

On the other hand, logarithmic decrement was adopted as well, Nakutis & Kaskonas in 2010 have calculated the damping ratio of a two-girder bridge using logarithmic decrement approach [6]. Chen et al. in 2020 have estimated the damping of long cables with viscous-shear dampers of the Sutong cable-stayed bridge located in China [7].

After employing all these mathematical functions in the developed LabVIEW code and treating the case of the beam in the steel cantilever beam on both LabVIEW and ARTeMIS for comparison purposes, the results showed a difference less than 5% which proves the accuracy and efficiency of LabVIEW. ”

[1] Structural Vibration Solutions. Available online: https://svibs.com/applications/operational-modal-analysis/ (accessed on 17/03/2023).

[2] Lin, C. W., & Yang, Y. B. Use of a passing vehicle to scan the fundamental bridge frequencies: An experimental verification. Engineering Structures 2005, 27(13), pp.1865-1878.

[3] Meng, X., Dodson, A. H., & Roberts, G. W. Detecting bridge dynamics with GPS and triaxial accelerometers. Engineering Structures 2007, 29(11), pp.3178-3184.

[4] Gonzalez, I., & Karoumi, R. Analysis of the annual variations in the dynamic behavior of a ballasted railway bridge using Hilbert transform. Engineering structures 2014, 60, pp.126-132.

[5] Chen, J., Xu, Y. L., & Zhang, R. C. Modal parameter identification of Tsing Ma suspension bridge under Typhoon Victor: EMD-HT method. Journal of Wind Engineering and Industrial Aerodynamics 2004, 92(10), pp.805-827.

[6] Nakutis, Ž., & Kaškonas, P. Bridge vibration logarithmic decrement estimation at the presence of amplitude beat. Measurement 2011, 44(2), pp.487-492.

[7] Chen, L., Di, F., Xu, Y., Sun, L., Xu, Y., & Wang, L. Multimode cable vibration control using a viscous‐shear damper: Case studies on the Sutong Bridge. Structural Control and Health Monitoring 2020, 27(6), p. e2536.

Point 2: While the validation of the system's accuracy is commendable, the paper could have included more discussion on potential sources of error and how they were addressed.

Response 2: Numerous source of error were eliminated by developing the LabVIEW code. They can be divided into two categories. First, the code itself is capable to eliminate them without the interference of the user such as the case of deleting the DC value from the raw acceleration-time data recorded by the accelerometers.

Therefore we have added this sentence in line 223:

“Figure 6 shows that the LabVIEW code calculates and eliminates the Mean (DC) value from the recorded acceleration in order to remove the offset from the raw acceleration time graph. This offset is considered as a potential source of error if it not eliminated.”  

Second, the developed LabVIEW has saved the user by not asking for the following input data before running the code.

Among them we can site drawing the geometry incorrectly or forgetting to add a structural element especially if the studied structure is considered complicated. Not forgetting the error that can happen if the user makes a mistake in incorrectly specifying the locus of each sensor mounted on the structure.

We have made sure to include these comments in the same paragraph that we have added for the sake for the first comment you have mentioned.

Point 3:  It would be beneficial if the paper could have included a more comprehensive evaluation of the system's performance under different conditions.

Response 3: The developed LabVIEW code was tested in the experiment of a cantilever beam with a steel plate cros-section reported in this research paper. But prior to writing this research, we have tested it in a case study of a real time pedestrian bridge used in a previous paper published by the same authors, which was published in the Journal of Civil Engineering and Construction in 2022. As it showed efficiency and accuracy of the developped code.

The pedestrian bridge is presented in the paper below:

El Dahr, R., Lignos, X., Papavieros, S. and Vayas, I. Design and Validation of an Accurate Low-Cost Data Acquisition System for Structural Health Monitoring of a Pedestrian Bridge. Journal of Civil Engineering and Construction 202211(3), pp.113-126.

We have seen that adopting the developed LabVIEW code for such complicated structures has saved the user by not asking for the following input data before running the code and have saved him from commiting several errors.

Among them we can site drawing the geometry incorrectly or forgetting to add a structural element especially if the studied structure is considered complicated, or even if the user forget to specify the location of each adopted sensor. Overall, the system performance under the condition of adopting such a structure did not contribute in any difference in the performance of the structure. As the accuracy of the results was validated with MATLAB code.

Reviewer 3 Report

The manuscript can be accepted in its current form.

Author Response

Response to Reviewer 3 Comments

We would like to thank you for taking the time to review our manuscript and for recommending it. We appreciate the opportunity you gave us to publish our research study in your reputable journal.

Reviewer 4 Report

The manuscript reports the development and testing of a LabVIEW software for structural health monitoring of a mechanical device based on vibrational measurements.
From a previously recorded data set, the software determines the natural frequencies and damping of a cantilever using LabVIEW built-in functions for fast Fourier transformation and Hilbert transformation. It is shown by the authors that the LabVIEW virtual instrument yields results comparable to commercial ARTeMIS software for the same purpose.

Since structural health monitoring is important in many technical applications, the reported results are relevant and fit into the field of 'Modelling'.

I recommend publication after consideration of a view minor points:

Line 16: Acronym BDI is not explained, most likely Bridge Diagnostic Inc.
Line 179: If possible, references for LabVIEW and ARTeMIS projects should be added.
Paragraph starting line 182:
The authors stress the advantage of LabVIEW having a "graphical interface, primarily adopted for data acquisition", thus shortening the "computational burden",  and this software would "overperform different DAQ techniques".
Can the authors provide some benchmarking information between the LabVIEW Vi on the one hand and the ARTeMIS software on the other hand? The reader might be interested in execution times and their dependency on sampling rates and data set volume. Performance issues might be important in cases where the data acquisition and processing using FFT/Hilbert transformation has to be handled more or less in real time on the same machine. I am not so sure that LabVIEW would outperform compiled source code using optimized math libraries. 

Line 240: ".. in order to tun it into .. " should probably mean ".. in order to convert it into .."
Line 323: For the manual calculation of the natural frequencies (section 9.1) the authors should explain the quantities E (most likely Young's modulus) and so on, for clarity.
Line 422: " .. was mount on .." should perhaps mean " .. was mounted on ..".

Author Response

Response to Reviewer 4 Comments

We would like to thank you for recommending this research study, and for your valuable remarks that we took into consideration for fulfilling the needed criteria in order to publish it in your reputable journal.

Point 1: Line 16: Acronym BDI is not explained, most likely Bridge Diagnostic Inc.

Response 1: we have added the acronym to be the following: “Bridge Diagnostic Inc. (BDI)”.

Point 2: Line 179: If possible, references for LabVIEW and ARTeMIS projects should be added.

Response 2: we have added the following paragraph in order to refer to assessed by the aforementioned codes:

“Both programs were adopted to assess large projects. For instance, LabVIEW was implemented by Li et al., in 2006, for developing the code to gather the signals recorded by sensors mounted on the Shandong Binzhou Yellow River highway bridge located in China for structural health monitoring purposes [1]. Same for ARTeMIS, the ambient assessment of the Vasco da Gama bridge located in Portugal was analysed on it during the design validation stage. Moreover, it was adopted in monitoring smaller bridge projects, the S101 highway bridge of Austria, in order to assess the damage and degradation over time bought by environmental influences [2].”

[1] Li, H., Ou, J., Zhao, X., Zhou, W., Li, H., Zhou, Z. and Yang, Y. Structural health monitoring system for the     Shandong Binzhou Yellow River highway bridge. Computer‐Aided Civil and Infrastructure Engineering 2006, 21(4), pp.306-317.

[2] Structural Vibration Solutions. Available online: https://svibs.com/applications/operational-modal-analysis/ (accessed on 17/03/2023).

Point 3: Paragraph starting line 182:
The authors stress the advantage of LabVIEW having a "graphical interface, primarily adopted for data acquisition", thus shortening the "computational burden",  and this software would "overperform different DAQ techniques".
Can the authors provide some benchmarking information between the LabVIEW Vi on the one hand and the ARTeMIS software on the other hand? The reader might be interested in execution times and their dependency on sampling rates and data set volume. Performance issues might be important in cases where the data acquisition and processing using FFT/Hilbert transformation has to be handled more or less in real time on the same machine. I am not so sure that LabVIEW would outperform compiled source code using optimised mathematical libraries. 

Response 3: ARTeMIS and LabVIEW are both softwares programs commonly used for data acquisition and analysis in the field of vibrational testing.

ARTeMIS is considered a reference code, as it was always implemented and tested by numerous sophisticated projects. However, we do think that the developed LabVIEW code still makes a significant contribution to the field. Both softwares differ in their approach to uploading data.

For that reason, we have established a comparison between them. In our revised manuscript, we will provide a more detailed discussion on benchmarking information regarding the uploaded data to both programs prior to post processing estimation and measurements calculation, in addition after running, the accuracy of the adopted mathematical functions was checked:

“Both programs demand the upload of the numerical data recorded by the accelerometers mounted on the structure, along with specifying the sensor’s sampling rate before running the codes.

It is noted that LabVIEW will not require from the user to draw the geometry of the structure neither to specify the location of each sensor mounted on the structure. Whereas ARTeMIS will request the drawing of the structure, specification of the locus of each sensor and allocation of each uploaded data with the specific directory (x,y,z) as the adopted accelerometers are triaxial.

The simplicity of the input required for LabVIEW will protect the user from committing any error and will save time. Moreover, it does not require to readjust the code when working on different projects. The user will upload the recorded measurements by the sensors and will only specify the sampling rate.

Once the code runs, LabVIEW will provide the calculated measurements from natural frequency, damping and displacement for each accelerometer, whereas ARTeMIS will show the response given by the most amplified signal.

The developed LabVIEW program relies mainly on mathematical functions such as Fast Fourier Transform (FFT) or Power Spectral Density (PSD) for natural frequency calculation, and Hilbert Transform (HT) as well as logarithmic decrement method for damping calculation. As these methods are considered the most suitable techniques since they were adopted in many projects and research papers.

For instance, FFT was adopted by Lin & Yang, in 2005 for estimating the natural frequency of the Da-Wu-Lun bridge located in Taiwan [3]. Mengh et al., in 2007 have employed FFT technique to calculate the natural frequency of the pedestrian Wilford Bridge located over River Trent in Nottingham [4].

By employing the hamming spectral window operator, FFT enables the determination of a power spectral density.

Same for the damping calculation, on one hand this research study has adopted the HT, as it was the most adopted by other researchers. Gonzalez & Karoumi in 2014 have calculated the damping of a bridge situated on the northern part of Sweden using HT [5]. Chen et al., in 2004 adopted a new emerged empirical mode decomposition (EMD) technique along with HT (EMD-HT) to calculate the damping of Tsing Ma suspension bridge located in Hong Kong [6].

On the other hand, logarithmic decrement was adopted as well, Nakutis & Kaskonas in 2010 have calculated the damping ratio of a two-girder bridge using logarithmic decrement approach [7]. Chen et al. in 2020 have estimated the damping of long cables with viscous-shear dampers of the Sutong cable-stayed bridge located in China [8].

After employing all these mathematical functions in the developed LabVIEW code and treating the case of the beam in the steel cantilever beam on both LabVIEW and ARTeMIS for comparison purposes, and the results showed a difference less than 5% which proves the accuracy and efficiency of LabVIEW. ”

[3] Lin, C. W., & Yang, Y. B. Use of a passing vehicle to scan the fundamental bridge frequencies: An experimental verification. Engineering Structures 2005, 27(13), pp.1865-1878.

[4] Meng, X., Dodson, A. H., & Roberts, G. W. Detecting bridge dynamics with GPS and triaxial accelerometers. Engineering Structures 2007, 29(11), pp.3178-3184.

[5] Gonzalez, I., & Karoumi, R. Analysis of the annual variations in the dynamic behavior of a ballasted railway bridge using Hilbert transform. Engineering structures 2014, 60, pp.126-132.

[6] Chen, J., Xu, Y. L., & Zhang, R. C. Modal parameter identification of Tsing Ma suspension bridge under Typhoon Victor: EMD-HT method. Journal of Wind Engineering and Industrial Aerodynamics 2004, 92(10), pp.805-827.

[7] Nakutis, Ž., & Kaškonas, P. Bridge vibration logarithmic decrement estimation at the presence of amplitude beat. Measurement 2011, 44(2), pp.487-492.

[8] Chen, L., Di, F., Xu, Y., Sun, L., Xu, Y., & Wang, L. Multimode cable vibration control using a viscous‐shear damper: Case studies on the Sutong Bridge. Structural Control and Health Monitoring 2020, 27(6), p. e2536.

Point 4: Line 240: ".. in order to tun it into .. " should probably mean ".. in order to convert it into .."

Response 4: we have corrected it.

Point 5: Line 323: For the manual calculation of the natural frequencies (section 9.1) the authors should explain the quantities E (most likely Young's modulus) and so on, for clarity.

Response 5: For steel: The Young's modulus E = 2x1011 N/m2           

                       The unit weight ρ = 7850 Kg/m3                        

                         b = 0.08 m, h = 0.006 m, L = 1.86 m

                         The area: A = b x h = 0.00048 m2

                         The moment of inertia: I = b h3 = 1.44x10-9 m4

 For cantilever beam the angular frequency ωn =  , for n (number of modes) = 1, 2, 3.

 Where  = 1.875,  according to [‎80].

 Therefore, ω1 = 8.88 rad/s, ω2 = 55.68 rad/s, ω3 = 155.92 rad/s

 The natural frequency f n =  , therefore f 1 = 1.413 Hz, f 2 = 8.861 Hz, f 3 = 24.815 Hz.

Point 6: Line 422: " .. was mount on .." should perhaps mean " .. was mounted on ..".

Response 6: we have corrected it.

As for the three criteria that you suggested to improve:

Are the conclusions supported by the results?

- In our research study we have taken appropriate measures to ensure that the results are clearly presented and well-supported in the conclusion. We made sure to present the results in both text and table format, which makes it more accessible for the reader to understand the findings. Moreover, the inclusion of percentage error between the results estimated by manual calculation, LabVIEW, and ARTeMIS can also provide valuable information about the accuracy of the different methods used.

Are all the cited references relevant to the research?

- We made sure that every reference was relevant to one of the scientific aspects discussed in this research and it is mandatory to site them just to avoid plagiarism or not give credits to previous scientific work.

Is the research design appropriate?

- Indeed, we have improved the research design in the revised manuscript as we added the description of the developed code benchmark and compare it with ARTeMIS, moreover , we have provided a more detailed discussion on benchmarking information regarding the uploaded data to both programs prior to post processing estimation and measurements calculation, in addition after running, the accuracy of the adopted mathematical functions was checked. Therefore we have covered any confusion the reader can get through to understand how each of the programs operates in terms of mathematical functions to calculate the measurements, and choose what is best for him according to the ease of implementation and the computational burden.

Furthermore, we have added how Numerous source of error were eliminated by developing the LabVIEW code. As they can be divided into two categories. First, the code itself is capable to eliminate them without the interference of the user such as the case of deleting the DC value from the raw acceleration-time data recorded by the accelerometers.

Therefore we have added this sentence in line 223:

“Figure 6 shows that the LabVIEW code calculates and eliminates the Mean (DC) value from the recorded acceleration in order to remove the offset from the raw acceleration time graph. This offset is considered as a potential source of error if it not eliminated.” 

Second, the developed LabVIEW has saved the user by not asking for the following input data before running the code.

Among them we can site drawing the geometry incorrectly or forgetting to add a structural element especially if the studied structure is considered complicated. Not forgetting the error that can happen if the user makes a mistake in incorrectly specifying the locus of each sensor mounted on the structure.

We have made sure to include these comments in the same paragraph that we have added for the sake for the third comment you have mentioned.

Round 2

Reviewer 1 Report

Innovation was the main concern of the paper. In terms of knowledge contribution, it is still weak in the revised version. It will be more appropriate to have included, as accompanying material, a more detailed description of the code design and development and share the sample code for the community to test their own data as a ‘repeatability’ evaluation.

Author Response

We would like to thank you for taking the time to review our manuscript. We appreciate your feedback and understand your concern regarding the novelty of our developed LabVIEW code.

Point 1: Innovation was the main concern of the paper. In terms of knowledge contribution, it is still weak in the revised version. It will be more appropriate to have included, as accompanying material, a more detailed description of the code design and development and share the sample code for the community to test their own data as repeatability evaluation.

Response 1: We would like to clarify that we have created a LabVIEW code that is capable of performing all the required integrations benefiting only from the mathematical functions accessible on LabVIEW platform. We made sure to present the developed operational technique in a very simple and detailed way for the user to understand how each evaluated measurement is calculated.

We have also implemented a technique to drop out the mean value in order to eliminate the effect of the offset on the integration results. This is important, as the offset can cause the integration to go monotonically up or down and generate wrong outcome and evaluation.

Generally, while conducting the same evaluation process on ARTeMIS and MATLAB, we have manually eliminated the mean value from the recorded acceleration time domain before uploading it as a (.txt) file.

In addition, we have also included a Butterworth filter to improve the accuracy of the results. The purpose of these features is to accurately calculate the eigenfrequencies, damping, and displacement from acceleration time domain data.

Moreover, for the displacement and damping ratio calculation, we have included the option of extracting the data, so that the user can validate that the integrations are starting at zero acceleration, and not at any position.

Our research can benefit the user from understanding how every computed measurement is estimated in terms of science and in terms of mathematical functions, the aspect that lacks in previous studies. And if he desires to compute the assessment by adopting our code, then the sequence of operations adopted in the system itself will define how every measurement is calculated.

We are sorry not to be able to respond to the, unusual might be said, reviewer’s request to share our code. However, we added, in section 6 a straight forward evaluation technique with all the mathematical functions adopted that allow the calculation of the eigenfrequencies, the damping ratio and the displacement, as shown in the attached Word file, and in the updated revised manuscript.

We hope that this information clarifies our methodology and addresses your concerns. If you have any further questions or comments, please do not hesitate to let us know.

Thank you again for your feedback and for considering our work for publication.

Round 3

Reviewer 1 Report

Thank you for addressing the comments and for revising the paper.

Just one more minor suggestion, if the figures can be presented in vector format instead of bitmap format, it would be helpful. E.g. in the case of readers wanting to repeat/use your proposed approach in LabVIEW.

Author Response

We would like to thank you for taking the time to review our manuscript. We appreciate your feedback and understand your concern regarding the novelty of our developed LabVIEW code.

Point 1: If the figures can be presented in vector format instead of bitmap format, it would be helpful. E.g. in the case of readers wanting to repeat/use your proposed approach in LabVIEW.

Response 1: We made sure to change all the figures to vector format (.svg).

Thank you again for your feedback and for considering our work for publication.
